# Light-Trapping Electrode for the Efficiency Enhancement of Bifacial Perovskite Solar Cells

**DOI:** 10.3390/nano12183210

**Published:** 2022-09-15

**Authors:** Anna A. Obraztsova, Daniele Barettin, Aleksandra D. Furasova, Pavel M. Voroshilov, Matthias Auf der Maur, Andrea Orsini, Sergey V. Makarov

**Affiliations:** 1School of Physics and Engineering, ITMO University, St. Petersburg 197101, Russia; 2Department of Electronic Engineering, Università Niccoló Cusano, 00133 Rome, Italy; 3Department of Electronic Engineering, University of Rome ‘Tor Vergata’, Via del Politecnico 1, 00133 Rome, Italy; 4Harbin Engineering University, Harbin 150001, China; 5Qingdao Innovation and Development Center of Harbin Engineering University, Qingdao 266000, China

**Keywords:** light trapping, perovskite solar cells, transparent conducting electrode, dielectric nanospheres

## Abstract

Antireflection and light-trapping coatings are important parts of photovoltaic architectures, which enable the reduction of parasitic optical losses, and therefore increase the power conversion efficiency (PCE). Here, we propose a novel approach to enhance the efficiency of perovskite solar cells using a light-trapping electrode (LTE) with non-reciprocal optical transmission, consisting of a perforated metal film covered with a densely packed array of nanospheres. Our LTE combines charge collection and light trapping, and it can replace classical transparent conducting oxides (TCOs) such as ITO or FTO, providing better optical transmission and conductivity. One of the most promising applications of our original LTE is the optimization of efficient bifacial perovskite solar cells. We demonstrate that with our LTE, the short-circuit current density and fill factor are improved for both front and back illumination of the solar cells. Thus, we observe an 11% improvement in the light absorption for the monofacial PSCs, and a 15% for the bifacial PSCs. The best theoretical results of efficiency for our PSCs are 27.9% (monofacial) and 33.4% (bifacial). Our study opens new prospects for the further efficiency enhancement for perovskite solar cells.

## 1. Introduction

Organic–inorganic lead halide perovskite solar cells (PSCs) are considered an up-and-coming substitute for well-known silicon solar cells. Perovskite materials possess excellent optical and electrical characteristics, such as a high light-absorption coefficient, long carrier lifetime, low exciton-binding energy and ambipolar transmission. Due to these vast advantages in the present photovoltaic (PV) industry, the use of PSCs for solar cells attracts tremendous research interest.

The simplest PSC structure receives light on a single side (monofacial), and it is generally provided via a front electrode based on transparent conducting oxides, such as FTO and ITO of an electron transport layer, a photoactive layer (perovskite), a hole transport layer, and non-transmitting metallic contact. Such solar cells exhibit energy conversion efficiency (PCE) up to 25.7% [1,2]. There are many strategies and special designs aiming to improve the PCE of a PSC. First of all is the realization of PSC structures with transparent electrodes at both sides, collecting light on both the device surfaces (bifacial). The bottom transparent conductive contact allows the reflected radiation to be harvested, i.e., albedo, thereby increasing the photon flux used to generate energy. A 10% efficiency improvement compared with monofacial cells has been demonstrated in commercial bifacial PV devices without significantly increasing production costs [3]. The idea of bifacial solar cells was first explored by A. Luque et al. [4], who explored concentrators and showed that using bifacial cells increased the concentration gain. Since then, different concepts concerning the fabrication of bifacial designs have been investigated [5,6,7,8].

Because of this great potential, it is highly important to analyze new designs of bifacial structures for future PSC deployment. Between all, the development of light-trapping electrodes (LTE) to overcome the high optical losses presents great potential both for monofacial and bifacial PSCs. The fundamental targets of light-trapping strategies are to minimize the incident light reflection, enhance the light absorption, and alter the optical response of the device for specific applications [9,10,11,12]. Various light-trapping structures and materials have been explored. Antireflective coatings on the outer side of solar cell structure are one of the most effective ways to improve the light absorption [13]. Daxue Du et al. explored nanostructures of various shapes for antireflection in organic–inorganic hybrid PSCs. They showed that the efficiency of their antireflective PSCs increased by 8.6% compared to the planar reference cell [14]. Other approaches to enhance light harvesting use photonic crystals [15], plasmonic nanostructures [16,17], random scatterers [18,19], microlenses [20,21,22,23,24,25], and nanostructures as nanowires [26,27,28], nanocones [29,30,31], nanorods [32,33,34], nanopillars [35,36,37], nanowells [38,39,40], triangular or pyramid structure [41,42,43,44,45], and nanospheres [46,47,48,49,50].

In this work, we propose a LTE structure based on silica nanospheres integrated over a MAPbI${}_{3}$-based perovskite solar cell. Thanks to the accurate sizing of the proposed structure, multiphysics theoretical simulations of the optimized bifacial architecture predicts a PCE as high as 33.4% without the need to increase the active material thickness.

## 2. Design and Methods

Thin-film PSCs are one of the best candidates for low-cost photovoltaic production, with minimal usage of active materials and simple device manufacturing. However, they have certain energy-loss mechanisms: low light absorption at the edge of the conduction band due to the restricted thickness of perovskite can decrease the generation rate and significantly reduce the efficiency of PSCs. The dominant optical loss mechanisms are reflection and parasitic absorption. These losses arise due to significant parasitic absorption of transparent conducting oxides (ITO, FTO), leads to a reduced photocurrent and, as a result, a decrease in PCE. Moreover, due to the high transparency of such electrodes and their refractive index values (n≈1.8), similar to a photoactive perovskite, unabsorbed photons can easily escape the device volume, making conventional PSCs less efficient than cells with metallic top contact. To reduce photon loss in ITO-based devices, a layer of titanium dioxide (TiO${}_{2}$) is used, which greatly improves the light absorption [51]. However, the electron transport layer TiO${}_{2}$ is still highly transparent, and its *n* value is not enough to significantly reduce the optical losses of solar cells. These devices also have reflection losses in the entire operating wavelength range, which reduce their PCE. Therefore, we focused on the enhancement of the optical absorption of the perovskite layer and reduced the reflection on the top layer of the device. To this end, we introduced a SiO${}_{2}$ array at the metallic contact with a light-focusing effect, which can possess additional functions such as self-cleaning and water repellent properties [52,53].

### 2.1. The Proposed LTE-Structure

In our work, we introduce a new way to improve the efficiency of PSCs through a significant reduction in their optical losses. The method relies on a light-trapping electrode structure (LTE structure), including a perforated metal electrode and densely packed dielectric nanospheres. In general, the production of regular arrays of silica nanospheres, which we proposed as light-trapping electrode technology, is a standard procedure among nanotechnology fabrication techniques, and is often used as a sacrificial mask for catalytic metals nanopatterning. Silica nanospheres are provided by strong adhesion on oxide surfaces thanks to hydroxyl groups surface bonding [54,55,56]. In fact, a single layer of silica nanospheres is easily obtainable via spin coating. In our paper, we chose a regular array of silica nanospheres with 940 nm diameter (Figure 1) as an optimum point for the light-trapping structure. On the other hand, from Figure 1b, it is evident that the area of higher gain (red and deep red color) does not strictly depend on the sphere diameter, and it may be comfortably chosen from 950 nm up to 900 nm. This structure can serve as an alternative to traditional electrodes in semitransparent PSCs based on transparent conducting oxides, such as ITO or FTO. Replacing the ITO with a highly reflective material is a sensible strategy to keep the radiation in the structure. We use metals, especially Al and Au, whose superior optical properties allow better light retention in PSCs than ITO does [13]. Using numerical calculations, we optimized the size of the structure and quantified the result by the parameter Gain (Equation (Equation 3)). The results of the evaluation are presented in Figure 1b. Figure 1a demonstrates a schematic illustration of the proposed structure of a semitransparent PSC with LTE. Through accurate selection of the dielectric sphere diameter and the diameter of the holes, there is significant improvement in the generation rate. We show that the proposed LTE can increase the generation rate of PSCs and their PCE up to 8%, mainly due to an increase in a short-circuit current density and a fill factor.

We performed separate numerical calculations to determine the pure impact of the optical and electronic effects from LTE introduced in PSCs and to consider the changes of the cells’ optical and electronic characteristics, such as absorption in the active, short-circuit current density and fill factor.

### 2.2. Optical Properties of Nanospheres

Subwavelength dielectric spheres with low dissipative losses can provide a light-focusing effect, which makes them very attractive for thin-film photovoltaics. For spheres with a refractive index of 1<n<2, the focus is located outside the sphere, at a distance from its shadow-side surface. For particles with n≈2, the incident light is focused exactly on their shadow-side surface, and for dielectric spheres with n>2, the incident light is focused inside the particle. The geometric optics focus can be found via Snell’s law: $f=\frac{R_{s}}{2}\frac{n}{n-1}$ (where $R_{s}$ is the sphere radius, and *n* is the refractive index), the spot size $r=R_{s}\sqrt{\frac{{\left(4-n^{2}\right)}^{3}}{27n^{4}}}$, and the field enhancement at the focal point is [57]
(1)$$\frac{I_{max}}{I_{0}}\approx \frac{R_{s}^{2}}{r^{2}}=\frac{27n^{4}}{{\left(4-n^{2}\right)}^{3}}.$$

From these formulas, we can conclude that the field is enhanced near the outer edge of the particle. The enhancement value depends only on the refractive index. The scattering of nanospheres is characterized by the size parameter: (2)$$q=\frac{2\pi R_{s}n}{\lambda },$$
when λ is the wavelength. To select the optimal parameters for a light-harvesting semitransparent electrode, we relied on this theory. The size parameter for the efficient light trapping by a perovskite layer should be in the range from 1.8π to 4π, which is an estimation based on our numerous calculations for various variables of the system. However, to estimate the increase in efficiency for our LTE compared to the reference structure, we should also consider the incident radiation for the solar cell in question. For this purpose, we introduced the parameter “Gain”:(3)$$Gain=\frac{\int A_{LTE}\cdot AM1.5d\lambda }{\int A_{Reference}\cdot AM1.5d\lambda },$$
where $A_{LTE},A_{Reference}$ are the optical losses in the active layer for the LTE-structured cell and the reference structure, respectively, and AM1.5 is the the spectral irradiance of AM 1.5 G.

### 2.3. Optical Calculations

For optical modeling, we used Maxwell’s equations as a basis to describe the light–matter interaction and establish the spectral characteristics of absorbance, reflectance and transmittance.
(4)$$\nabla \times E=-\frac{\partial B}{\partial t},$$
(5)$$\nabla \times H=\frac{\partial D}{\partial t},$$
(6)B=μH,
(7)D=εE,
where *H* is the magnetic field, *D* is the electric displacement field, *E* is the electric field, *B* is the magnetic field, ε is the complex permittivity, and μ is the complex permeability. The optical characteristics of PSCs were obtained using the finite-difference frequency-domain (FDFD) algorithm, which is a rigorous electromagnetic calculation used to solve Maxwell’s equations. Here, we assumed that each absorbed photon with energy greater than the semiconductor bandgap can generate one electron–hole pair, and the photogeneration rate can be presented as: (8)$$G=\int \frac{\varepsilon_{0}\omega {\left|E\left(\lambda \right)\right|}^{2}Im\left(\varepsilon \right)PFD_{AM1.5}}{2\hbar \omega }d\lambda ,$$
where $\varepsilon_{0}$ is the dielectric constant, ω is the angular frequency of the incident light, λ is the wavelength, E(λ) is the electric field depending on the wavelength, Im(ε) is the imaginary part of the permittivity, and $PFD_{AM1.5}$ is the photon flux density under AM1.5 G conditions.

### 2.4. Electrical Calculations

The bandgaps, the valence band offsets of all the materials in our simulations, and the Fermi levels of the contacts are shown in Figure 2. The cathode and the anode contacts were modeled as selective Schottky contacts to simulate the presence of electron and hole transport layers.

We solved the transport of carriers using a drift-diffusion model, which couples the continuity and Poisson equations: $$\nabla \cdot j_{n}=\nabla \cdot \left(\mu_{n}n\nabla \phi_{n}\right)=-R+G,$$
$$\nabla \cdot j_{p}=\nabla \cdot \left(\mu_{p}p\nabla \phi_{p}\right)=R-G,$$
(9)∇·(ε∇φ−P)=ρ.

The first two equations are the continuity equations for the electron and hole currents, with *n* and *p* being the electron and hole densities, $\mu_{n}$ and $\mu_{p}$ the electron and hole mobility, $\phi_{n}$ and $\phi_{p}$ the electron and hole quasi Fermi levels and *G* and *R* generation and recombination rates, respectively.

The generation rate *G* is read from a data file, together with the coordinates of fields. A finite element mesh is created from the coordinates via a 2D or 3D triangulation, and the generation data are then interpolated linearly onto the simulation grid. The data file of the generation rate is directly derived from the optical simulation results.

The recombination *R* includes the Shockley–Reed–Hall $R_{SRH}$ and direct recombination $R_{DIR}$, which are given by
$$R_{SRH}=\frac{np-n_{i}^{2}}{\tau_{n}\left(p+p_{i}\right)+\tau_{p}\left(n+n_{i}\right)},$$
(10)$$R_{DIR}=k_{2}\left(np-n_{i}^{2}\right).$$

The recombination parameters $\tau_{n}$, $\tau_{p}$, and $k_{2}$ are as in [58]. Auger recombination has been neglected, since it is usually not relevant in PSCs.

In the Poisson equation, which is the last equation in (Equation 9), φ is the potential and ρ the total density, which includes free carrier densities, trap distributions, ionized donors, and acceptors. Finally, ε is the permittivity of the material and *P* is the polarization field.

The equations are solved using the finite elements method (FEM) using TiberCAD simulation software [59,60,61]. This model for PSCs is described in more detail in [58,62].

## 3. Results

### 3.1. Optical Analysis of R, T, A for the Monofacial Perovskite Solar Cell

PSCs require transparent conducting electrodes, which allow incident solar radiation to penetrate the structure without losing the ability to conduct an electrical current. However, the conventional indium tin oxide (ITO) electrodes suffer from complex manufacturing problems and also have high parasitic absorption. The latter significantly reduces the radiation flux in the active layer and the generation of electron–hole pairs. In this paper, we propose using the LTE structure as a transparent electrode instead of the FTO and ITO commonly used for semitransparent PSCs. As a common basis, we used a basic structure of a one-sided perovskite solar cell, as shown in Figure 3a. The reference structure consists of a 400 nm-thick FTO (indium doped tin oxide) deposited on a flat glass, followed by a 200 nm-thick TiO${}_{2}$, 300-nm thick CH${}_{3}$NH${}_{3}$PbI${}_{3}$, 200 nm-thick Spiro-OMeTAD, and 100-nm thick Au layer.

Figure 3a shows a schematic illustration of the new proposed structure, in which we introduced glass nanospheres located on a perforated metal substrate as a transparent electrode. We consider sunlight normally incident from the LTE side. According to the reported simulations in Figure 3b,c, there is light focusing at the sphere bottom and improved light harvesting in the active layer. Figure 3c shows the power flow for the LTE structure, calculated for a wavelength of 650 nm, where a characteristic spot of field enhancement is observed at the focus point. For the reference structure (Figure 3d), there are no characteristic spots of field enhancement in the perovskite layer, and this design has significant optical losses in the red wavelength range (Figure 3e). The power flow reaches the maximum in the FTO and TiO${}_{2}$ layers for the reference design, and decreases significantly in the active layer. In our proposed design, the maximum power is in the TiO${}_{2}$ layer, which is associated with a low coefficient of extinction, but the flux power in the active layer is much higher than in the reference design.

Regardless of the carrier loss during the charge transportation, the current is determined by the amount of light absorbed by the active layer, where the charge generation occurs. Therefore, we performed a numerical calculation of the optical parameters: the absorption spectrum in the active layer, parasitic absorption and reflection. We mainly focused on the optical losses in the wavelength range of 350–800 nm, which are determined by the bandgap of the perovskite material. As seen in Figure 3e, although the perovskite layer absorbs most of the incident sunlight, there is still a huge potential for increasing the generation rate in the reference PSC. The reflection loss and parasitic absorption are observed in the entire wavelength range, and they decrease the total optical efficiency by more than 20% (Table 1).

On the other hand, as shown in Figure 3b,e, the percentage of parasitic absorption for the proposed LTE structure at normal incidence of light is 6.8%, which is almost half the absorption in the reference structure. Moreover, this LTE reduces the reflection losses by 3%, and increases the active layer’s absorption by 9%. We also estimated the effective improvement with the Gain parameter to be 11% for our structure at normal incidence (Table 1).

### 3.2. Optical Analysis of R, T, A for the Bifacial Perovskite Solar Cell for the Front and the Back Side

In this work, we designed LTE structure for a bifacial perovskite solar cell with two transparent electrodes. On the one hand, transparent electrodes increase the efficiency of the solar cell by harvesting the reflected light (from the surface on which the electrode is located), and also allow the integration of the developed solar cells into the facades of houses, car windows, etc. However, due to high light transmission, the efficiency of photocurrent generation in the active layer may decrease. Therefore, we propose an LTE as an active contact in a bifacial perovskite solar cell that can keep the light from the front side and increase the light harvesting from the back side. Figure 4a shows a schematic illustration of the proposed structure, in which this perforated electrode is used as the bottom (located on the back side) contact. Optical analysis results for this design are shown in Figure 4b (for normal light incidence from the back side) and in Figure 4c (for normal light incidence from the front side). Figure 4d–f presents data for the reference cell with ITO material as a top contact: the schematic illustration, the results for the normal back side and front side light incidence. Here, the light absorption in the active layer is enhanced on the back side by 12.4%, and on the front side by 1.3% (Table 2). The absorption of light by the active layer provides the formation of electron–hole pairs, which is necessary for energy conversion and forms the efficiency of the solar cell. The increase in light absorption by the active layer results from a 16.5% decrease in reflection losses for the back side, and a 1.7% decrease in transmittance for the front side (Table 2). We estimated the efficiency increase for the back side of the solar cell is $Gain_{backside}=15\%$, and for the front side, $Gain_{frontside}=1\%$.

## 4. Incoherent Illumination

To calculate the photocurrent and efficiency of a solar cell, we use the absorption spectrum of incident light from a coherent source. However, sunlight is an incoherent source: incoming waves from the sun have a finite coherence time (finite spectrum width). In the absorption spectrum from a coherent source, one observes oscillations (Fabry–Perot type). On the contrary, for incoherent radiation, such oscillations disappear due to destructive interference effects. This effect can be seen in Figure 5. This picture shows the absorption spectrum in the active layer in both cases for the reference structure (the one-sided solar cell) and for our LTE structure. We calculated the incoherent efficiency of solar cells analytically. From the coherent one, we derived the product of the convolution with the function characterizing incoherent light (11, 12). This approach follows directly from Maxwell’s equations and is described in detail in [63]
(11)$$I\left(\omega \right)=\tau_{c}\sqrt{\frac{\ln2}{\pi^{3}}}e^{-\frac{\ln2}{\pi^{2}}\tau_{c}^{2}\omega^{2}},$$
where I(ω) is the incoherence function, $\tau_{c}$ is the coherence time, and ω is the cyclic frequency.
(12)$$A_{incoh}=I\left(\omega \right)\times A_{coh},$$
where $A_{incoh}$ is the incoherent absorption, and $A_{coh}$ is the coherent absorption.

According to the calculations, the parameter “Gain” did not change significantly for PSCs when they were illuminated by coherent and incoherent light, and therefore, for our structure, it is acceptable to use the coherent source.

## 5. Electrical Analysis of Monofacial and Bifacial Perovskite Solar Cells

The second part of calculations is the electrical characteristics for the proposed solar cell structure, required to verify that the integration of LTE structure can improve the main PV parameters of PSCs. Figure 6 shows the estimated current density vs. voltage plots for PSCs with the LTE, and for the reference structure for both considered designs (monofacial and bifacial). In the calculations, we assumed that the current density for the double-sided PSC is created from both front and back illumination of the structure, taking into account that the back side absorbs only 30% of the incident radiation [64]. Thus, due to the increase in the absorption coefficient in the active layer and reduction in optical losses, the short-circuit current increases. As we can see in Table 3, the short-circuit current increases from $17.6\frac{\mathrm{mA}}{{\mathrm{cm}}^{2}}$ to $18.5\frac{\mathrm{mA}}{{\mathrm{cm}}^{2}}$ for the monofacial PSCs, and from $26.5\frac{\mathrm{mA}}{{\mathrm{cm}}^{2}}$ to $27.4\frac{\mathrm{mA}}{{\mathrm{cm}}^{2}}$ for the bifacial PSCs. In turn, the open-circuit voltage remains the same.

Note that the metallic perforated electrode also has a significantly lower sheet resistance compared to the ITO-based electrode. For ITO, it is 8–9 $\frac{\Omega }{{\mathrm{cm}}^{2}}$, and for the perforated metal, it is 2–3 $\frac{\Omega }{{\mathrm{cm}}^{2}}$. It is known that the sheet resistance is proportional to the series resistance $\rho_{S}\left(\frac{\Omega }{{\mathrm{cm}}^{2}}\right)≅R_{S}\left(\Omega \right)$, where $\rho_{S}$ is the sheet resistance and $R_{S}$ is the series resistance [65]. Reducing the series resistance increases the fill factor and also affects the efficiency gain. $F_{S}=F_{0}\left(1-R_{S}\right)$, where $F_{0}$ is the fill factor excluding the series resistance and $F_{S}$ is the fill factor including the series resistance. When plotting the current density curve as a function of voltage, we obtained the curve without the series resistance using the method described above and reconstructed it with series resistance as $I_{S}=I_{S}\left(V-R_{S}I_{0}\right)$, where $I_{0}$ is the current density excluding series resistance and $I_{S}$ is the current density including series resistance. Thus, we obtained an increase in fill factor FF for the monofacial structure from 74.8% to 76%, and for the bifacial structure from 80.3% to 81.4% (Table 3). Thus, our proposed structure has excellent optical and electrical characteristics.

The efficiency of the one-sided and double-sided PSCs has increased up to 1.8% and 1.5% respectively.

## 6. Conclusions

In summary, we have demonstrated a novel approach for the optimization of current generation in PSCs by using a light-trapping structure that improves the absorption in the active layer and reduces the optical losses, such as reflection and parasitic absorption. We have modeled the optical and electrical performance of our design of PSCs and showed that our structure has excellent characteristics, enhancing the efficiency of light conversion and current generation. In addition, the perforated metal electrode has lower sheet resistance than ITO or FTO, which is an advantage for solar cell applications. We have observed 11% improvement in light absorption for the monofacial PSCs, and a 15% for the bifacial PSCs. The best theoretical results for our PSCs’ efficiency are 27.9% (monofacial) and 33.4% (bifacial). Thus, our electrode based on the light-trapping strategy is one of the best candidates to replace classic transparent conducting oxides such as ITO or FTO.

## Figures and Tables

**Figure 1 nanomaterials-12-03210-f001:**
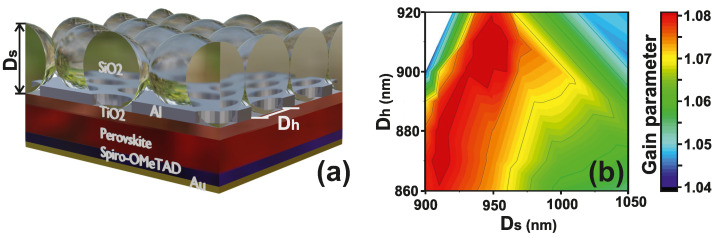
Optimization: (**a**) The scheme of the proposed light-trapping electrode structure. $D_{h}$ is the diameter of holes, $D_{s}$—diameter of spheres. (**b**) Gain parameter depends on the sphere diameter and hole diameter (calculated for λ= 400–800 nm).

**Figure 2 nanomaterials-12-03210-f002:**
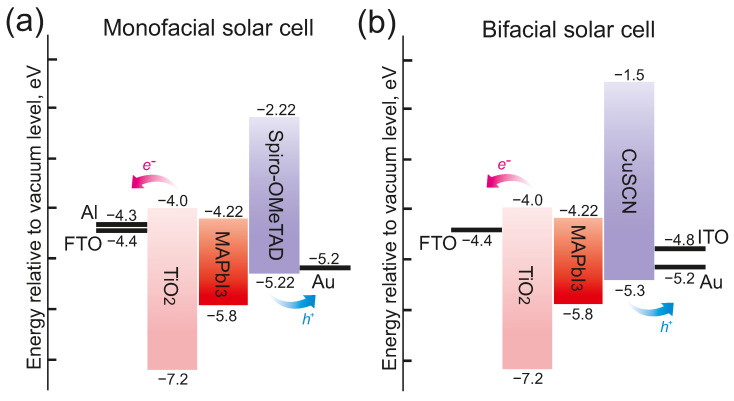
Band diagram of PSCs and the principle of charge separation: (**a**) Band diagram of monofacial PSCs. (**b**) Band diagram of bifacial PSCs.

**Figure 3 nanomaterials-12-03210-f003:**
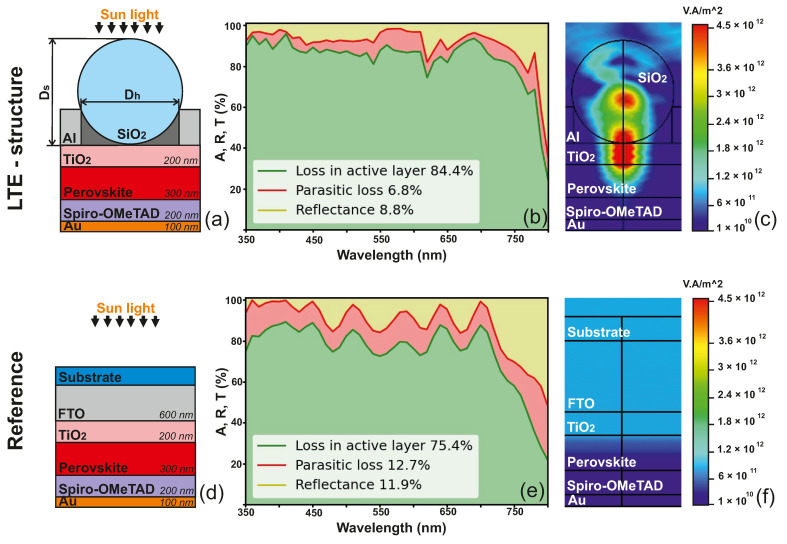
Optical efficiency analysis of monofacial PSCs (calculated for λ= 350–800 nm). (**a**) The scheme of light-trapping structure for monofacial PSC, $D_{h}$ is the diameter of holes ($D_{h}=$ 900 nm), $D_{s}$—diameter of spheres ($D_{s}=$ 940 nm), (**b**) spectral absorption, transmission, and reflection of the light-trapping structure for the monofacial PSC, (**c**) power flow of the light-trapping structure for the monofacial PSC at a wavelength of 650 nm with a nanosphere size 940 nm (TE—polarization), (**d**) scheme of the reference monofacial PSC, (**e**) spectral absorption, transmission, and reflection of the reference monofacial PSC, (**f**) power flow of the reference monofacial PSC at a wavelength of 650 nm with a nanosphere size 940 nm (TE—polarization).

**Figure 4 nanomaterials-12-03210-f004:**
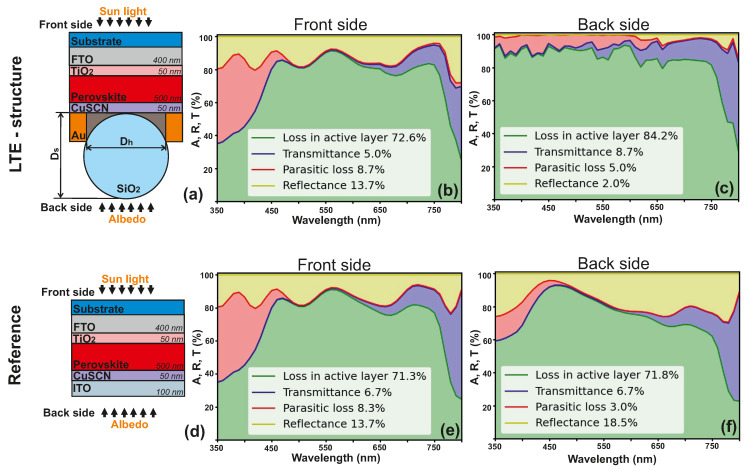
Optical efficiency analysis of bifacial PSCs calculated for λ= 350–800 nm. (**a**) Scheme of light trapping structure for the bifacial perovskite solar cell, $D_{h}$ is the diameter of holes ($D_{h}=$ 900 nm), and $D_{s}$—diameter of spheres ($D_{s}=$ 940 nm), (**b**) spectral absorption, transmission, and reflection for the front side illumination of the light trapping structure for the bifacial PSC, (**c**) spectral absorption, transmission, and reflection for the illumination of the back side of light-trapping structure for the bifacial PSC, (**d**) scheme of the reference bifacial PSC, (**e**) spectral absorption, transmission, and reflection for the front side illumination of the reference bifacial PSC, (**f**) spectral absorption, transmission, and reflection for the back side illumination of the reference bifacial PSC.

**Figure 5 nanomaterials-12-03210-f005:**
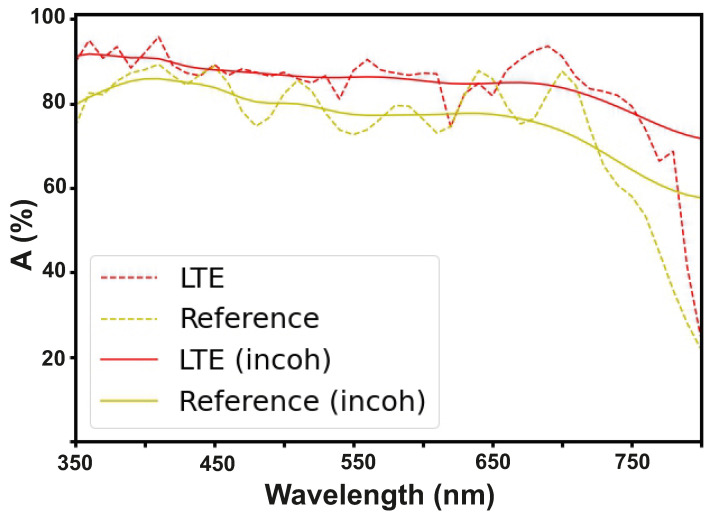
Incoherent illumination: Spectral absorption in active layer (calculated for λ= 350–800 nm).

**Figure 6 nanomaterials-12-03210-f006:**
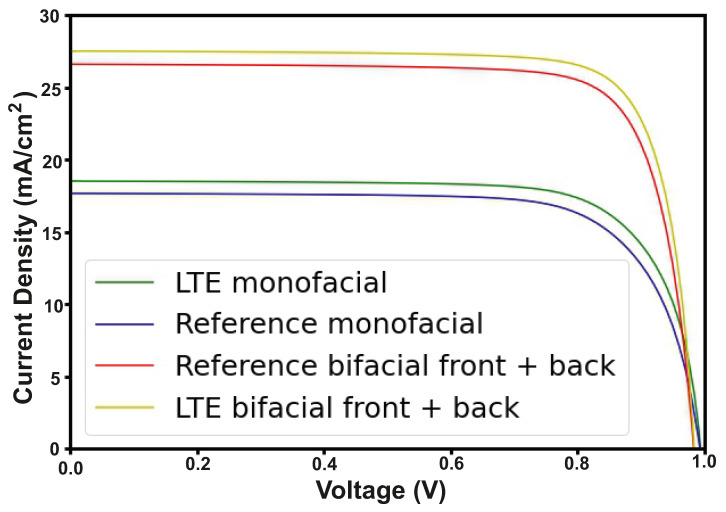
J-V curves for PSCs: monofacial with light-trapping structure(green), monofacial reference (blue), bifacial with light-trapping structure (yellow), bifacial reference (red).

**Table 1 nanomaterials-12-03210-t001:** Optical characteristics of monofacial perovskite solar cells.

Design	Loss in the Active Layer (%)	Parasitic Loss (%)	Reflectance (%)	Gain (%)
Monofacial reference PSC	75.4	12.7	11.9	
Monofacial PSC with LTE	84.4	12.7	11.9	11

**Table 2 nanomaterials-12-03210-t002:** Optical characteristics of bifacial perovskite solar cells.

Design	Loss in the Active Layer (%)	Transmittance (%)	Parasitic Loss (%)	Reflectance (%)	Gain (%)
Bifacial reference PSC (front side)	71.3	6.7	8.3	13.7	
Bifacial PSC with LTE (front side)	72.6	5	8.7	13.7	1
Bifacial reference PSC (back side)	71.8	6.7	3	18.5	
Bifacial PSC with LTE (back side)	84.2	8.7	5	2	15

**Table 3 nanomaterials-12-03210-t003:** Photovoltaic characteristics of monofacial and bifacial PSCs.

Design	$V_{\mathrm{OC}}\left(V\right)$	$J_{\mathrm{SC}}\left(\frac{\mathrm{mA}}{{\mathrm{cm}}^{2}}\right)$	FF(%)	PCE(%)
Monofacial reference PSC	0.99	17.6	74.8	26.1
Monofacial PSC with LTE	0.99	18.5	76.0	27.9
Bifacial reference PSC	0.98	26.5	80.3	31.9
Bifacial PSC with LTE	0.98	27.4	81.4	33.4

## Data Availability

The data presented in the current work are available on request from the corresponding authors and Appendix A.

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
