# Peer review of "Light-Trapping Electrode for the Efficiency Enhancement of Bifacial Perovskite Solar Cells"

_nanomaterials, 2022, doi:10.3390/nano12183210_

Round 1

Reviewer 1 Report

The authors use the light-trapping electrode (LTE) to replace classical transparent conducting oxides such as ITO or FTO; The results of numerical calculations show that the light-trapping electrode can provide better optical transmission and conductivity. Based on the LTE, the best theoretical PCE of MAPbI3-based PSCs are 27.9% (mono-facial) and 33.4% (bi-facial). Overall, this work represents a significant advance in reducing the optical losses of PSCs, which provides a reference method for improving the efficiency of the perovskite solar cells. I recommend acceptance after major revisions:

1. This manuscript lacks necessary references and supporting information, such as detailed fabrication methods for LTE and devices.

2. The traditional vacuum-treated gold or copper metal electrodes always have poor adhesion, so how is the adhesion of LTE?

3. Can LTE be used as transparent electrodes on both sides of the device?

4. Figure 3c and Figure 3f show the power flow of the device at a wavelength of 650 nm with a nanosphere size of 940 nm (TE – polarization). Please give more detailed discussions on these two figures.

5. Many calculations have been carried out in this manuscript. I suggest the authors give some tables to show the calculation results. 

Reviewer 2 Report

The authors report a ligh-trapping electrode (LTE) to replace classical transparent conducting oxides electrode in PSC devices fabricated by a perforated metal film covered with densely packed array of nanospheres. The LTE possesses better optical transmission and conductivity, acting as charge collection and light trapping, which is in favor of bifacial perovskite solar cells. By simulated calculation, an enhanced PCE of 33.4% for bifacial PSC can be achieved with increased short-circuit current density and fill factor attributed to the improvement of light absorption. Overall, The author should take into account the following major revisions:

1. The author should give a wider overview on the present scenario related to current trends in bifacial PSCs.

2. According to the calculation, device with LTE shows larger transmittance than ITO when illumination from back side, whereas, the transmittance decreases to lower than ITO with illumination from front side. Please explain the reversal of transmittance upon illumination from different side.

3. What about the transmittance of LTE itself as well as ITO in comparation?

4. In addition to simulated calculation, the author should give a real device using LTE and its power conversion efficiency, as well as the reference device for comparation.

5. The author claims the advantages of LTE in calculation, however, the preparation of such configuration (perforated metal film covered with desnsely packed array of nanospheres) is missed. Is such LTE actual available for PSC devices? Moreover, the morphology of the LTE such as SEM should be provided.

6. There are some grammatical errors in the manuscript. For example, on page 3, line 129: “we can significant improvement in the generation rate”. Please check and revise carefully.

Reviewer 3 Report

In this paperthe authors a novel approach for the optimization of current generation in PSCs by using a light-trapping structure that improves the absorption in the active layer and reduces the optical losses, such as reflection and parasitic absorption. But the manuscript is still needed a revision before published.

1、 As can be seen in figure 1, the gain parameter depends on the sphere diameter and hole diameter. Will the difference between the internal and external radius of the hole diameter affect the gain parameter?

2、 In figure 3, the thickness of each layer of perovskite in the experimental design is unreasonable. And the thicknesses in figure 3 and 4 do not correspond.

3、 In table 1, we can see the photovoltaic characteristics of monofacial and bifacial PSCs. Does Vos correspond to Voc ? Device parameters do not correspond to PCE, and the Jsc is wrong.

4、 The 1st paragraph of Design and methods: The sentence ‘These losses arise due to significant parasitic absorption of transparent conducting oxides (ITO, FTO), which leads to a reduced photocurrent and, as a result, a decrease in PCE.’ is abrupt and has no correlation with the previous content.

5、 Is antireflection coating a kind of light-trapping strategy? At the end of the 1st paragraph of Design and methods, it seems that antireflection coatings and light-trapping strategies are parallel strategies. In 2.1. Light trapping structuring, it seems that light-trapping strategies contain antireflection coatings.

Reviewer 4 Report

The manuscript demonstrated a novel strategy for the improvement of efficiency of PSCs by using a light-trapping structure. The method improved the absorption in the active layer and reduced the optical losses, and finally enhanced the efficiency of light conversion and current generation. The article is well organized, but there are a few minor problems that I would like the author to explain:

1.     Compared to the reference devices, the efficiency of the one-side and double-side PSCs is increased up to 1.8% and 1.5% respectively. Is this the best result and can it be further improved by structural optimization?

2.     The back electrode of the reference device is ITO, while the back electrode of light trapping structure PSC is Au. Why there are two different back electrodes used?

3.     In Figure 3, why the electrode of the reference device did not choose the same material for comparison?

Round 2

Reviewer 1 Report

This muanscirpt can be accepted now.

Reviewer 3 Report

no comments